# CaloDREAM —
# Detector Response Emulation via Attentive flow Matching

Luigi Favaro, Ayodele Ore, Sofia Palacios Schweitzer, and Tilman Plehn

Institut für Theoretische Physik, Universität Heidelberg, Germany

May 20, 2024

## Abstract

Detector simulations are an exciting application of modern generative networks. Their sparse high-dimensional data combined with the required precision poses a serious challenge. We show how combining Conditional Flow Matching with transformer elements allows us to simulate the detector phase space reliably. Namely, we use an autoregressive transformer to simulate the energy of each layer, and a vision transformer for the high-dimensional voxel distributions. We show how dimension reduction via latent diffusion allows us to train more efficiently and how diffusion networks can be evaluated faster with bespoke solvers. We showcase our framework, CaloDREAM, on datasets 2 and 3 of the CaloChallenge.

# 1 Introduction

Simulations are the way we compare theory predictions to LHC data, allowing us to draw conclusions about fundamental theory from complex scattering data [1,2]. The modular simulation chain starts from the hard interaction and progresses through particle decays, QCD radiation, hadronization, hadron decays, to the interaction of all particles with the detector. Currently, the last step is turning into a bottleneck in speed and precision. Generating calorimeter showers with GEANT4 [3–5], based on first principles, requires a large fraction of the computing budget. Without significant progress, this simulation step will be the limiting factor for all analyses at the HL-LHC.

Modern machine learning is transforming the way we simulate LHC data [6]. In the past few years we have seen successful applications to all steps in the simulation chain [2], phase space integration [7–17]; parton showers [18–25]; hadronization [26–29]; detector simulations [30–63], and end-to-end event generation [64–71], including inverse simulations [72–82] and simulation-based inference [83–85]. While these new concepts and tools have the potential to transform LHC simulations, we need to ensure that these networks and their technical strengths can be be understood. This is the only way that we can systematically improve the LHC simulation chain [69, 86–89], without endangering the key role it plays in, essentially, every LHC analysis. Most notably, we have to ensure that LHC simulations remain first-principle and are not replaced by data-driven modelling. This means that for now we always assume that networks used in LHC simulations are trained on simulations and controlled by comparing to simulations.

In this paper, we will apply cutting-edge generative networks to calorimeter shower simulations. The high-dimensional phase spaces of calorimeter showers are a challenge to the established normalizing flows or INNs [90], and different variants of diffusion networks appear to be the better-suited architecture [56]. This is in spite of the fact that diffusion networks are, typically, slower in the forward generation and do not allow for an efficient likelihood extraction. In addition to showing that these networks are able to simulate sparse phase space signals like calorimeter showers, we will explore which phase space dimensionalities we can describe with full-dimensional latent spaces and how a dimension-reduced latent representation affects the network performance.

Given the GEANT4 benchmark presented in Sec. 2, we will see that a factorized approach is most promising. In Sec. 3 we first introduce a Conditional Flow Matching (CFM) network combined with an autoregressive transformer to learn the layer energies. Next, we combine it with a 3-dimensional vision transformer to learn the shower shapes. This combination can be trained on datasets 2 and 3 of the CaloChallenge to generate high-fidelity calorimeter showers. In this application the step from dataset 2 to dataset 3 motivates a switch from full-dimensional voxel representations to a dimension-reduced latent space [48]. In Sec. 4 we study, in some detail, how the full-dimension generative network encodes the calorimeter shower information for both datasets. To alleviate the computational challenges, we also show how a lower-dimensional latent representation helps us describe high-dimensional data like the Calo Challenge dataset 3 and how the CFM networks can sample more efficiently. For quantitative benchmarking of the learned phase space distribution we employ a learned classifier test, indicating that the network precision for both datasets is at the per-cent level and the loss in precision from a reduced latent space is controlled, including its only failure mode, which are the sparsity distributions.

## 2   Data and preprocessing

To benchmark our new network architectures, we use dataset 2 (DS2) [91] and dataset 3 (DS3) [92] of the Calo Challenge at ML4Jets 2022 [93]. Each set consists of 200k GEANT4 [3] positron showers: 100k for training/validation and 100k for testing. Showers are simulated over a log-uniform incident energy range

$$E_{\text{inc}} = 10^3 \dots 10^6 \text{ MeV} \,. \tag{1}$$

The physical detector has a cylindrical geometry with alternating layers of absorber and active material, altogether 90 layers. The voxelization following Ref. [93] combines an active layer and an absorber layer resulting in 45 concentric cylindrical layers.

   The particle originating the shower always enters at the (0,0,0) location and defines the $z$-axis of the coordinate system. The number of readout cells per layer is defined in a polar coordinate system and it is different for DS2 and DS3. DS2 has a total of 6480 voxels: 144 voxels per layer, each divided into 16 angular and 9 radial bins. DS3 has a much higher granularity with 40500 total voxels, where the number of layers is unchanged but the angular and radial binning is 50×18. Both datasets have a threshold of 15.15 keV. While this is an unrealistic cut for practical applications, it provides a useful challenge to high-dimensional generative networks covering a wide energy range.

### Preprocessing

We improve our training by including a series of preprocessing steps, similar to previous studies [47,49,55,56,90]. We split information on the deposited energy from its distribution over voxels by introducing energy ratios [39]

$$u_0 = \frac{\sum_i E_i}{f \, E_{\text{inc}}} \qquad \text{and} \qquad u_i = \frac{E_i}{\sum_{j \geq i} E_j} \quad i = 1, \dots, 44 \,, \tag{2}$$

where $E_i$ refers to the total energy deposited in layer $i$, and $f \in \mathbb{R}$ is a scale factor. The number of $u$-variables matches the number of layers. With these variables extracted from a given shower, we are free to normalize the voxel values by the energy of their corresponding layer without losing any information. This definition is analytically invertible, imposes energy conservation, and ensures that the normalized voxels and each $u_{i>0}$ are always in the range [0, 1]. However, due to the calibration of the detector response caused by the inactive material, $u_0$ can have values larger than 1. We set $f = 2.85$ in Eq.(2), to rescale $u_0 \in [0,1]$. All networks are conditioned on $E_{\text{inc}}$. This quantity is passed to the network after a log transformation and a rescaling into the unit interval.

   To train the autoencoders used for dimensionality reduction we do not use any additional preprocessing steps. For the setup using the full input space, we apply a logit transformation regularized by the parameter $\alpha$ which rescales each input voxel $x$,

$$\begin{aligned} x_\alpha &= (1 - 2\alpha)x + \alpha \in [\alpha, 1 - \alpha] \qquad \text{with} \quad \alpha = 10^{-6} \\ x' &= \log \frac{x_\alpha}{1 - x_\alpha} \,. \end{aligned} \tag{3}$$

Finally, we calculate the mean and the standard deviation of the training dataset and standardize each feature. The postprocessing includes an additional step that rescales the sum of the generated voxels to ensure the correct normalization in each layer.

# 3 CaloDREAM

In CaloDREAM*, we employ two generative networks, one energy network and one shape network [39]. The energy network learns the energy-ratio features conditioned on the incident energy, $p(u_i|E_{inc})$. The shape network learns the conditional distribution for the voxels, $p(x|E_{inc}, u)$. The two networks are trained independently, but are linked in the generative process. Specifically, to sample showers given an incident energy, we follow the chain

$$u_i \sim p_\phi(u_i|E_{inc})$$
$$x \sim p_\theta(x|E_{inc}, u). \tag{4}$$

In this notation $\phi$ stands for the weights in the energy network and $\theta$ for the weights in the shape network. Although the number of calorimeter layers is consistent across DS2 and DS3 and the underlying showers are the same, we train separate energy networks for each dataset. The incident energy is always sampled from the known distribution in the datasets, as in Eq.(1).

## 3.1 Energy network — Transfusion

Both of our generative networks use the Conditional Flow Matching architecture [94]. It starts with the ordinary differential equation (ODE)

$$\frac{dx(t)}{dt} = v(x(t), t) \qquad \text{with} \qquad x \in \mathbb{R}^d, \tag{5}$$

and a velocity field $v(x(t), t) \in \mathbb{R}^d$. This time evolution can be related to the underlying density through the continuity equation

$$\frac{\partial p(x, t)}{\partial t} + \nabla_x [p(x, t)v(x, t)] = 0. \tag{6}$$

The velocity field transforms the density $p(x, t)$ such that

$$p(x, t) \to \begin{cases} \mathcal{N}(x; 0, 1) & t \to 0 \\ p_{data}(x) & t \to 1. \end{cases} \tag{7}$$

We can estimate the velocity field with $v_\phi(x(t), t)$. In this case we can sample the data distribution from Gaussian random numbers, tracing the trajectory using any ODE solver. Defining the training trajectories to be linear, the velocity network is optimized using a simple MSE loss

$$\mathcal{L}_{CFM} = \left\langle \left[ v_\phi((1-t)\epsilon + tx, t) - (x - \epsilon) \right]^2 \right\rangle_{t \sim U(0,1), \epsilon \sim \mathcal{N}, x \sim p_{data}}. \tag{8}$$

Conditional probability distributions can be learned by allowing $v_\phi$ to depend on additional inputs.

For the energy network, we exploit the causal nature of the energy deposition in layers using an autoregressive transfusion architecture [85], as visualized in Fig 1. We start by embedding $E_{inc}$ as our one-dimensional condition and the $u$-vector. For the $u$, this is done by concatenating a one-hot encoded position vector and zero-padding. These embeddings are passed to the encoder and decoder of a transformer, respectively. For the one-dimensional condition the encoder's self-attention reduces to a trivial $1 \times 1$ matrix. For the decoder we

---

*The code used for this paper is publicly avaiable at https://github.com/heidelberg-hepml/calo_dreamer

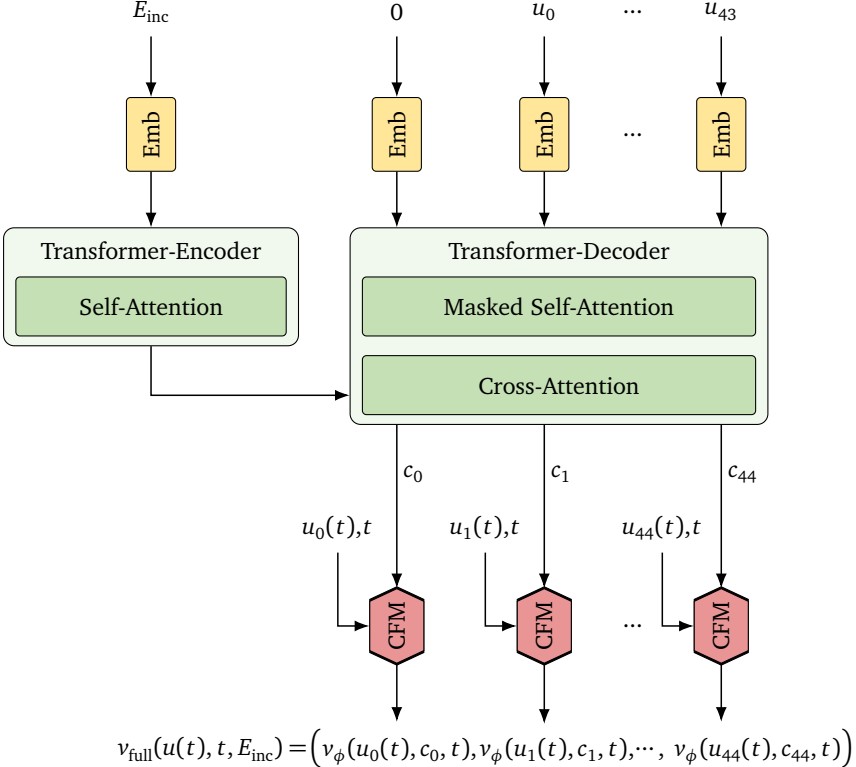

Figure 1: Schematic diagram of the autoregressive Transfusion network [85] used in our energy network.

mask our self-attention with an upper triangle matrix, to keep the autoregressive conditioning. Afterward, we apply a cross-attention between the encoder and decoder outputs. The transformer outputs the vectors $c_0, \ldots, c_{44}$, encoding the incident energy and previous energy ratios,

$$c_i = \begin{cases} c_i(u_0, \ldots, u_{i-1}, E_{\text{inc}}) & i > 0 \\ c_i(E_{\text{inc}}) & i = 0 \, . \end{cases} \tag{9}$$

For generation, we use a single dense CFM network $v_\phi$, with the inputs time $t$, embedding $c_i$, and the point on the diffusion trajectory $u_i(t)$. This network is evaluated 45 times to predict each component of the velocity field individually,

$$v_{\text{full}}(u(t), t, E_{\text{inc}}) = \left( v_\phi(u_0(t), c_0, t), \ldots, v_\phi(u_{44}(t), c_{44}, t) \right) \tag{10}$$

During training, we can evaluate the contribution of each $u_i$ to the loss in parallel, whereas sampling requires us to iteratively predict the $u_i$ layer by layer. The hyperparameters of the transfusion network are given in Tab. 1.

## 3.2 Shape network — Vision Transformer

For the shape network, we use a 3-dimensional vision transformer (ViT) to learn the conditional velocity field $v_\theta(x(t), t, E_{\text{inc}}, u)$. The architecture is inspired by Ref [95] and illustrated in Fig. 2. It divides the calorimeter into non-overlapping groups of voxels, so-called patches, which are embedded using a shared linear layer and passed to a sequence of transformer

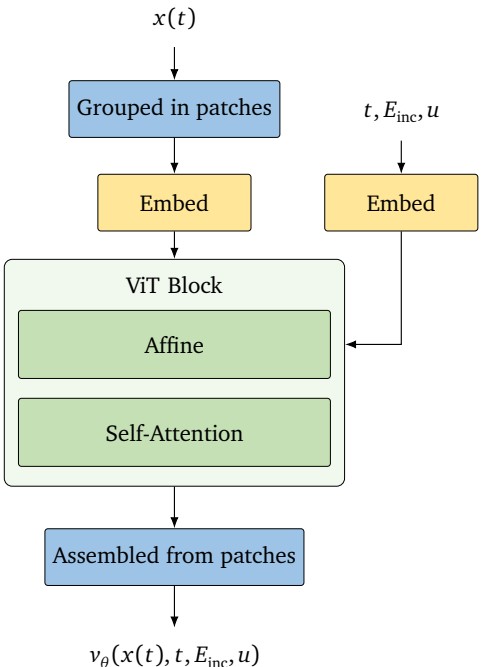

Figure 2: Schematic diagram of the vision transformer (ViT) [95] used in our shape network.

blocks. Each block consists of a multi-headed self-attention and a dense network that transforms the patch features. To break the permutation symmetry among patches, we add a learnable position encoding to the patch embeddings prior to the first attention block. After the last block, a linear layer projects the processed patch features into the original patch dimensions, where each entry represents a diffusion velocity. Finally, the patches are reassembled into the calorimeter shape.

The network uses a joint embedding for the conditional inputs, $t, E_{\mathrm{inc}}$ and $u$. The time and energy coordinates are embedded with separate dense networks, then summed into a single condition vector. The attention blocks incorporate this condition via affine transformations with shift and scale $a, b \in \mathbb{R}$ and an additional rescaling factor $\gamma \in \mathbb{R}$ learned by dense layers. These are applied within each block, and also to the final projection layer. Concretely, the operation inside the ViT block is summarized by

$$
\begin{aligned}
x_{\mathrm{h}} &= x + \gamma_{\mathrm{h}} g_{\mathrm{h}}(a_{\mathrm{h}} x + b_{\mathrm{h}}), \\
x_{\mathrm{l}} &= x_{\mathrm{h}} + \gamma_{\mathrm{l}} g_{\mathrm{l}}(a_{\mathrm{l}} x_{\mathrm{h}} + b_{\mathrm{l}}),
\end{aligned}
\tag{11}
$$

where $g_{\mathrm{h}}$ is the multi-head self-attention step and $g_{\mathrm{l}}$ is the fully connected transformation. The hyperparameters of our transformer are given in Tab. 2.

The scalability of this architecture is closely tied to the choice of patching. On the one hand, small patches result in high-dimensional attention matrices. While this gives a more expressive network, the large number of operations can become a limitation for highly-granular calorimeters. Conversely, a large patch size compresses many voxels into one object, implying a faster forward pass but at the expense of sample quality. In this case, an expanded embedding dimension is needed to keep the network flexibility fixed.

Usually, we train Bayesian versions [96] of all our generative networks, including calorimeter showers [90]. In this study, the networks learning DS2 and DS3 are so heavy in terms of operations, that an increase by a factor two, to learn an uncertainty map over phase space, surpasses our typical training cost of 40 hours on a cutting-edge NVIDIA H100 GPU. In prin-

ciple, Bayesian versions of all networks used in this study can be built and used to quantify limitations, for instance related to a lack of training data.

## 3.3 Latent diffusion

As the calorimeter granularity is increased from DS2 to DS3, the computational requirements to train a network on the full voxel space also increase considerably due to the larger number of patches. This motivates a study of how the naive scaling may be avoided by a lower-dimensional latent representation. Starting from the detector geometry, a voxel-based representation of a shower defines a grid with fixed size and stores the deposited energy in each voxel. This means a highly granular voxelization will produce a large fraction of zero voxels, but the showers should define a lower-dimensional manifold of the original phase space. Such a manifold can then be learned by an autoencoder [48, 90, 97].

We train a variational autoencoder with learnable parameters $\psi$. The encoder outputs a latent parameter pair $(\mu, \sigma)$, which defines the latent variable $r = \mu + z \cdot \sigma$ with $z \sim \mathcal{N}(0, 1)$. The encoder distribution represents the phase space distributions over $x$ through $p_\psi(r|x, u)$. For simplicity, in the following we drop the energy dependence in the encoder and decoder distributions. After sampling the latent variable, we minimize the learned likelihood of a Bernoulli decoder $p_\psi(x|r)$ represented by the reconstruction loss

$$\mathcal{L}_{\text{VAE}} = \left\langle -\log p_\psi(x|r) \right\rangle_{x \sim p_{\text{data}}, r \sim p_\psi(r|x)} + \beta \left\langle D_{\text{KL}}[p_\psi(r|x), \mathcal{N}(0, 1)] \right\rangle_{x \sim p_{\text{data}}}. \quad (12)$$

This choice of likelihood is possible since our preprocessing normalizes voxels into the range $[0, 1]$. The reconstruction quality achieved in the autoencoder training places an upper bound on the quality of a generative model trained in the corresponding latent space.

The KL-divergence term, with unit-Gaussian prior and a small weight $\beta = 10^{-6}$, is a regularization rather than a condition for a tractable latent space. It encourages a smooth latent space, over which we train the generative network. Especially for DS3, an autoencoder trained

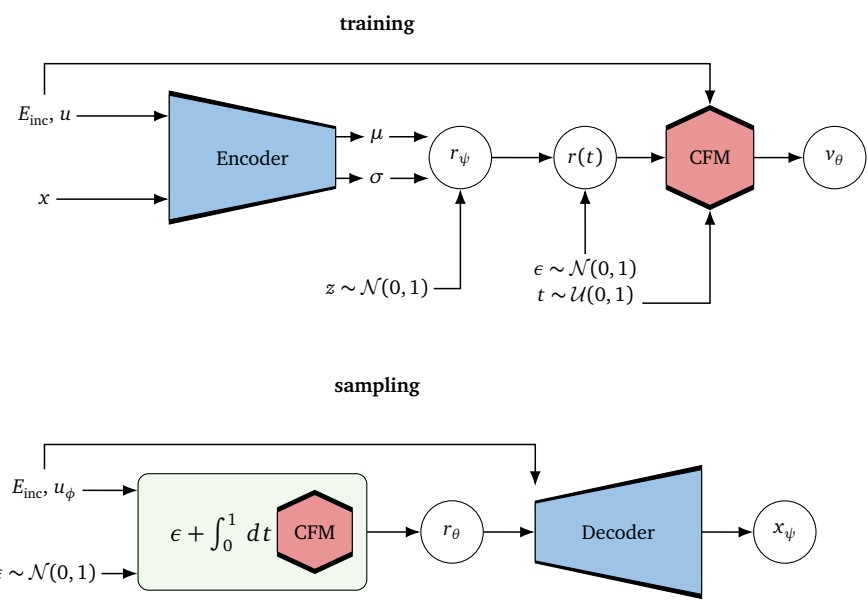

Figure 3: Training (upper) and sampling (lower) with the latent diffusion network, using a variational autoencoder.

without KL-regularization produces a sparse latent space with features mapped over several orders of magnitude.

The VAE consists of a series of convolutions, the last of which downsamples the data. This structure is mirrored in the decoder using ConvTranspose operations. As always, the energy conditions are encoded in a separate network and passed to the encoder and decoder. For a compressed latent space the ratio between the dimensionality of $x$ and $r$ defines the reduction factor $F$. Rather than estimating the dimensionality of the datasets, we use a moderate, fixed reduction factor $F \simeq 2.5$ and a bottleneck with two channels. We provide more details on the autoencoder training in App. A.2.

The trained autoencoder is used as a pre- and postprocessing step for the CFM as illustrated in Fig. 3. Given the trained encoder distribution $p_\psi(r|x)$ the velocity field $v(r(t), t)$ imposes the boundary conditions

$$p(r, t) \rightarrow \begin{cases} \mathcal{N}(r; 0, 1) & t \rightarrow 0 \\ p_\psi(r|x) & t \rightarrow 1, \ x \sim p_{\text{data}}. \end{cases} \tag{13}$$

The expensive sampling then uses the lower-dimensional latent space and yields samples $r$ from the learned manifold. Finally, the phase space configurations are provided by the the deterministic decoder $D_\psi(r)$. Here we summarize the sampling procedure, including the energy dependence, as three sequential steps:

$$\begin{aligned} u &\sim p_\phi(u|E_{\text{inc}}) \\ r &\sim p_\theta(r, 1|u, E_{\text{inc}}) \\ x &= D_\psi(r, u, E_{\text{inc}}) \end{aligned} \tag{14}$$

All network hyperparameters and the main training parameters are given in App. A.1.

## 3.4 Bespoke samplers

A potential drawback of CFM networks is their slower sampling than, for instance, normalizing flows with coupling layers [90] which stems from the numerical integration of the ODE in Eq.(5). Depending on the complexity of the target distribution, a standard ODE solver requires $\mathcal{O}(100)$ steps to achieve high-fidelity samples, each consisting of at least one forward pass of the neural network.

One method to overcome this slow inference is distillation [55, 61, 98, 99], which aims to predict the sampling trajectory at only a handful of intermediate points, or even at the terminus in a single step. This requires fine-tuning the network weights using additional training time, in some cases even additional training data. Further, since the weights of the network itself are updated, consistency is not strictly guaranteed and we can end up sampling from a different distribution than was originally learned.

An alternative approach is to keep the network fixed and consider alternative structures for the ODE solver. Reference [100] provides a comparison of various training-free solvers in the context of calorimeter simulations. While training-free approaches are the least costly, they are not task-specific and therefore unlikely to be optimal. However, there exists trainable family of ODE solvers that can be optimized to a given vector field $v_\theta$ without excessive additional training [101, 102]. Such Bespoke Non-Stationary (BNS) solvers parameterize the steps along the flow trajectory. Starting from an initial state $x_0$, and a time discretization $0 = t_0 < t_i < t_N = 1$, the $i^{\text{th}}$ integration step is

$$x_{i+1} = a_i x_0 + b_i \cdot V_i \quad \text{with} \quad a_i \in \mathbb{R}, \ b_i \in \mathbb{R}^{i+1}$$
$$V_i = [v_\theta(x_0, t_0), \cdots, v_\theta(x_i, t_i)] \in \mathbb{R}^{(i+1)\times d}, \tag{15}$$

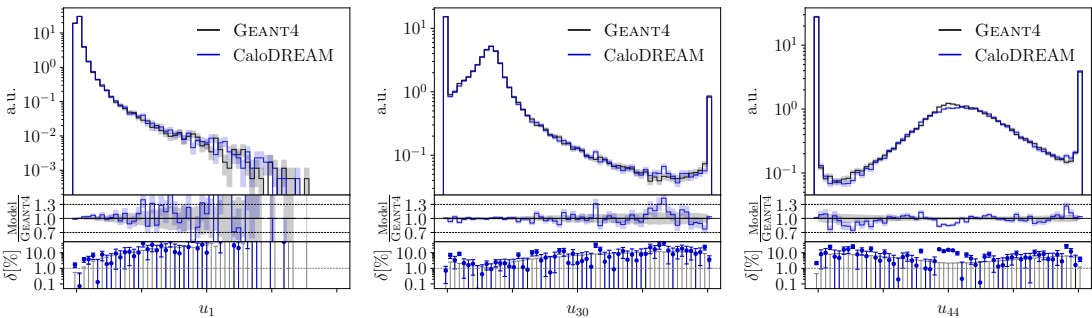

Figure 4: Distributions of selected $u$-features in DS2 from the CaloDREAM energy network (blue) compared to truth (grey). The error bars in all feature distributions in this paper show the statistics of the respective datasets.

where we again suppress the energy dependence of $v_\theta$. By appropriately caching the velocities, each step requires just one evaluation of the network. Including the $t_i$ not fixed by the boundary conditions, an $N$-step BNS solver has a total of $N(N+5)/2+1$ learnable parameters, which is typically orders of magnitude fewer than the network $v_\theta$. Therefore, optimizing the solver generally requires a fraction of the computation time needed to train the vector field itself. Non-stationary solvers encompass a large family of ODE solvers, including the Runge-Kutta (RK) methods. Euler's method, i.e. first order RK, corresponds to taking $a_i = 1$ and $b_{ij} = 1/N$.

Bespoke solvers can be trained by comparing the bespoke trajectory to a precisely-computed reference $x_{\text{ref}}(t)$, given an initial state $x_0$ sampled from the CFM latent distribution. Here we define two options. First, the global truncation error measures the deviation between the final states of the solvers

$$\mathcal{L}_{\text{GTE}} = \left\langle [x_{\text{ref}}(1) - x_N]^2 \right\rangle_{x_0 \sim \mathcal{N}} , \tag{16}$$

where $x_N$ is computed by iterating Eq.(15) starting from $x_0$. The local truncation error instead measures the discrepancy at each step,

$$\mathcal{L}_{\text{LTE}} = \left\langle \sum_{i=0}^{N-1} \left[ x_{\text{ref}}(t_{i+1}) - (a_i x_0 + b_i \cdot V_{\text{ref},i}) \right]^2 \right\rangle_{x_0 \sim \mathcal{N}} , \tag{17}$$

where $V_{\text{ref},i}$ is defined as in Eq.(15), but with velocities evaluated on the reference trajectory.

Although we use CFM for both our shape and energy networks, we only study BNS solvers for the shape network. For training a BNS solver, we initialize it to the Euler method. At each iteration, we sample an $x_0$ batch from the unit Gaussian and a batch of conditions from the energy network. A precise solver is then used to generate the reference trajectory $x_{\text{ref}}(t)$ which enters the loss. Note that the shape model parameters $\theta$ are frozen during training. The complete list of hyperparameters are given in App. A.1.

## 4 Results

### 4.1 Layer energies

In Fig. 4 we compare samples generated from the energy network with the truth for a selection of normalized layer energies $u_i$. The transfusion network indeed generates high-quality

distributions, with errors comparable to the statistical uncertainties in the test data. The distributions for $u_{i>40}$ are the most difficult to model, since the majority of showers lie in the sharp peaks at zero or one. These are zero-width peaks corresponding to showers that end at the given layer, leading to a one, or end before or skip the layer, leading to a zero.

We find that our autoregressive setup is particularly effective in faithfully mapping regions close to these peaks. As a quantitative performance measure, we train a classifier to distinguish the $u$'s defined by our energy network from the GEANT4 truth, obtaining AUC scores around 0.51. The comparison in terms of layer energy is shown in Fig. 5. The factorization procedure allows us to use the same energy network for the ViT and the laViT, effectively generating statistically-identical layer energy distributions.

## 4.2 DS2 showers

Given the learned layer energies, we use the shape networks described in Sec. 3.2 to generate the actual calorimeter showers over the voxels. First, we evaluate the distribution of energy depositions per layer by looking at shape observables, like the center of energy of the shower and its width in the $\phi$ and $\eta$ directions,

$$\langle \xi \rangle = \frac{\xi \cdot x}{\sum_i x_i}$$

$$\sigma_{\langle \xi \rangle} = \sqrt{\frac{\xi^2 \cdot x}{\sum_i x_i} - \langle \xi \rangle^2} \qquad \text{for} \qquad \xi \in \{\eta, \phi\} \,. \tag{18}$$

Here $x_i$ is the energy deposition in a single voxel and the sum runs over the voxels in a layer.

In the first row of Fig. 5 we compare a set of layer-wise distributions from the networks trained in the full space and in the latent representation to the test data truth. We start with the energy deposited in layer 20, where for $E_{20} > 10$ MeV the full-dimensional vision transformer (ViT) as well as the latent-diffusion counterpart (laViT) agree with the truth at the level of a few per-cent, as expected. Towards smaller energies we see a missing feature in both networks. Also in the two other shown distributions the ViT and laViT agree with each other and deviate from GEANT4 only in regions with statistically limited training data.

The second row of Fig. 5 shows example distributions probing the combination of layers. In addition to the layer-wise shower shapes, we calculate the mean shower depth weighted by the energy deposition in each of the $N$ layers for slices in the radial direction,

$$d_{r_j} = \frac{\sum_i^N k_i E_{i,r_j}}{E_{\text{tot},j}} \qquad r_j \in \{0, \dots, |r|\} \,. \tag{19}$$

Here $E_{i,r_j}$ is the average energy deposition in slice $r_j$, and $E_{\text{tot},j}$ is the total energy deposition in the selected slice. Slices in the angular direction are less interesting to calculate due to the rotational invariance of the showers. This observable highlights a small deviation for both networks from the reference for showers with maximum depth of five layers not captured by the layer-wise high-level features.

Also combining layer-wise information, we show the total energy deposited in the calorimeter $E_{\text{tot}}$ normalized by the incident energy and the full voxel distribution across the entire calorimeter $E_{\text{voxel}}$. The total shower energy relative to the incident energy is reproduced very well by both networks since this information is coming from the energy network. However for the voxel energies only the full-dimensional network captures the low-energy regime, whereas the latent model overestimates this regime and in turn shifts down the prediction for larger

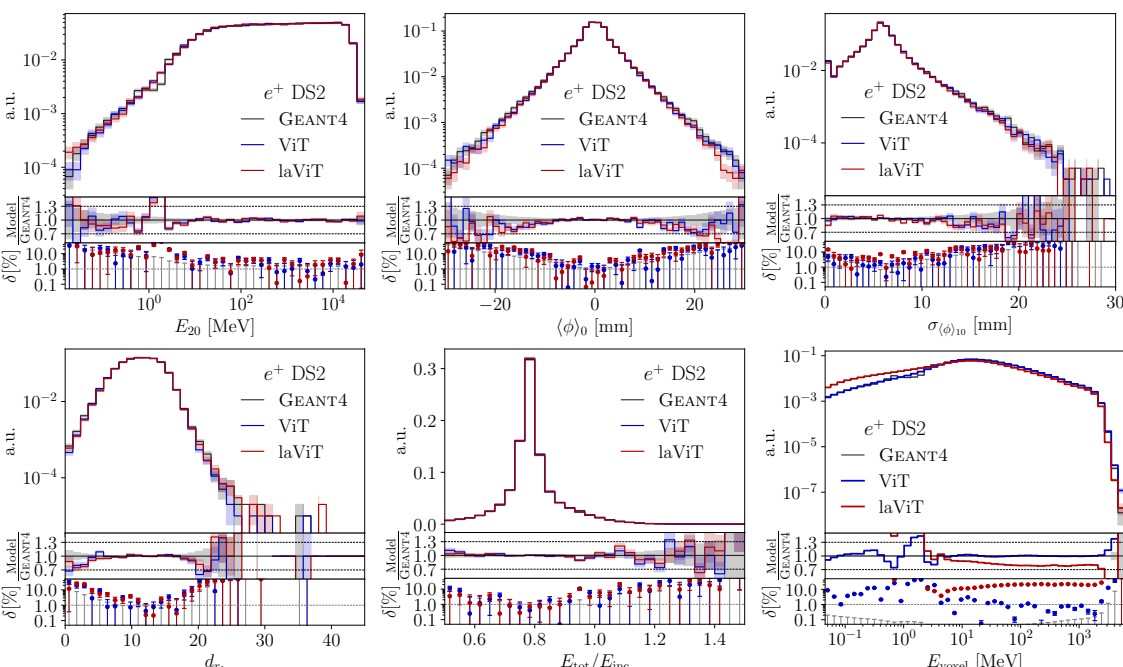

Figure 5: Selection of high-level features for DS2. The first row shows features for individual layers, the second row the combination of layers.

energies because of the normalization of the curve. This is the only noteworthy shortcoming of the laViT compared to the ViT that we find.

Following up on the problem raised by the last panel in Fig. 5, we focus on the (latent) description with low-energy voxels. In Fig. 6 we again compare the two network predictions with the truth, but applying an additional threshold cut of

$$E_{\text{voxel}} > 1 \text{ MeV} \,. \tag{20}$$

After this cut, the agreement of the laViT prediction with the full ViT and the truth improves significantly. We checked that this cut has only a limited impact on the total energy deposition $E_{\text{tot}}$. Slight deviations are limited to the threshold region $E_{\text{voxel}} \lesssim 5$ GeV. The reason can be seen in the sparsity distributions for instance of layer 10, $\lambda_{10}$. The laViT networks generates a sizeable number of showers with energy depositions everywhere, leading to a peak at zero sparsity. This failure mode is already present in the autoencoder reconstruction as described in App. A.2. Because of their low energy, these contributions do not affect the other high-level

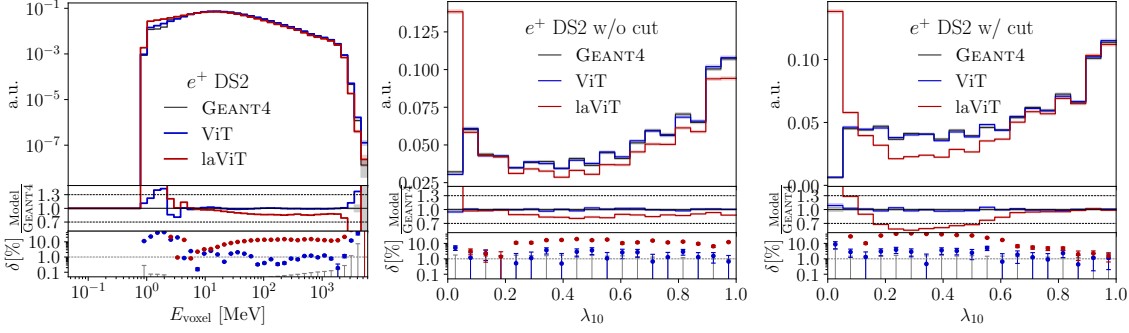

Figure 6: Effect of an additional threshold $E > 1$ MeV on DS2; we show the shower energy and the sparsities without and with threshold cut.

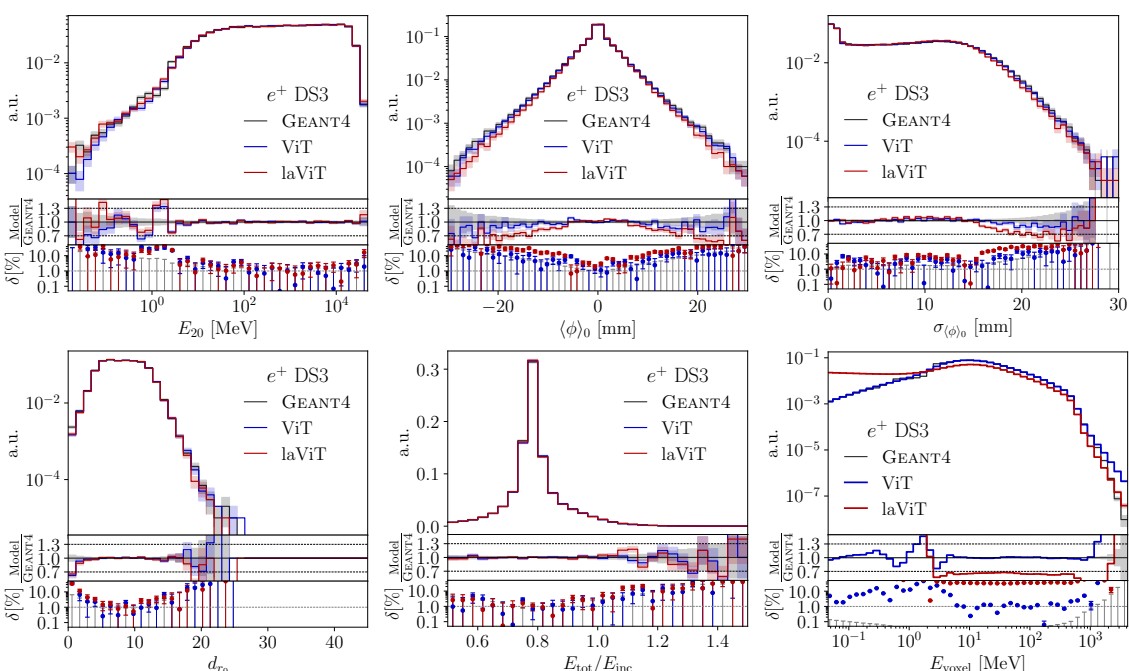

Figure 7: Selection of high-level features for DS3. The first row shows features for individual layers, the second row the combination of layers. All features correspond to the DS2 results shown in Fig. 5.

observables or the learned physics patterns of the showers.

## 4.3 DS3 showers

The same analysis done for DS2 in Sec. 4.3 we now repeat for DS3. This means we study the same shower energies and shower shapes, but from 40500 instead of 6480 voxels. A target phase space of such large dimension is atypical for most LHC applications, and the key question is whether the precision-generative networks developed for lower-dimensional phase spaces also give the necessary precision for high-dimensional phase spaces. As a matter of fact, we know that this is not the case for standard normalizing flows or INNs [90], where the architectures have to be modified significantly to cope with higher resolution.

In Fig. 7 we again show a set of layer-wise features in the first row, observing extremely

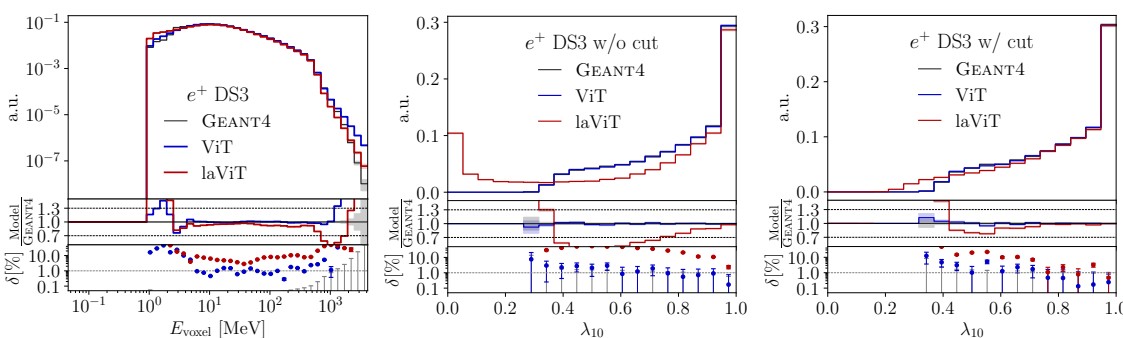

Figure 8: Effect of an additional threshold $E > 1$ MeV on DS3; we show the shower energy and the sparsities without and with threshold cut. All features correspond to the DS2 results shown in Fig. 6.

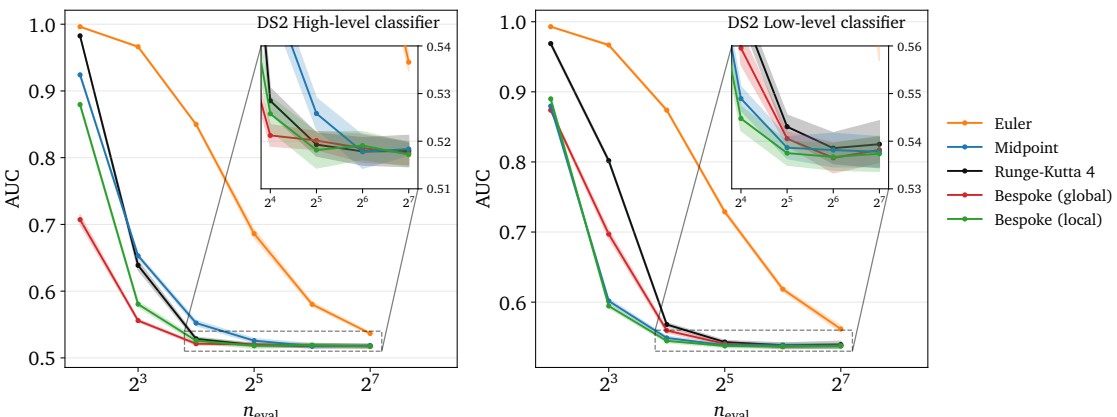

Figure 9: High-level (left) and low-level (right) classifier AUC scores on DS2 as a function of the number of function evaluations $n_{\mathrm{eval}}$ for various ODE solvers. Samples of 100k reference and generated showers are used to train the classifier. Errors bands are taken as the standard deviation over 10 runs.

mild differences to the DS2 results. Only the shower shapes from the laViT suffer slightly in regions with too little training data. For the multi-layer features in the second row, we also find the same results as for DS2, including the challenge in describing voxels with $E_{\mathrm{voxel}} \lesssim 3$ GeV.

Understanding and targeting this challenge, we again show the voxel energy distribution and the sparsity after the threshold cut $E_{\mathrm{voxel}} > 1$ MeV in Fig. 8. For DS3 it turns out that after applying this cut the description of DS3 through the laViT network is excellent. The reason for this is two-fold. Given the low energy bound we can reproduce with the latent model, a cut larger than this threshold completely adjusts the sparsity up to a specific value by removing the additional energy deposition of the latent model and the noisy components of GEANT4. For both DS2 and DS3 the cut fixes the sparsity in $\lambda_{10}$ up to $\lambda_{10} \gtrsim 0.7$. However, for DS3 this is done by moving the peak at zero, while for DS2 the mass is moved from the intermediate sparsity. This second difference comes from the dimensionalities of the two datasets, where the fixed reduction factor has a stronger impact on DS2 due to the larger information loss in the bottleneck.

## 4.4 Sampling efficiency

To demonstrate the performance of bespoke samplers, we compare the quality of showers produced by various solvers in terms of classifier tests. Classifiers trained to distinguish generated and true samples are an effective diagnostic tool since they capture failure modes in high-order correlations that are hidden in simple high-level distributions. As we will see in the following section, the phase space distribution of classifier scores can be used to search for and identify such failure modes. In this section, we only use the AUC as a simple, one-dimensional quality measure. The high-level classifier uses the layer-wise features but since we want sensitivity also to voxel-level correlations, we train a classifier on the low-level phase space as defined by the original voxels.

For our comparison, we include three standard fixed-steps solvers: the Euler, Midpoint, and Runge-Kutta 4 methods. We also consider bespoke non-stationary solvers using either the global, Eq.(16), or local, Eq.(17) truncation error as described in Sec. 3.4. Using each solver, we generate 100k showers from the DS2 ViT shape network. We train classifiers to distinguish these samples from the GEANT4 reference set using the standard CaloChallenge pipeline. In Fig. 9, we plot the high-level (left) and low-level (right) AUC scores against the

number of function evaluations $n_{\text{eval}}$ for each solver. Note that the Midpoint and RK4 methods respectively use 2 and 4 function evaluations per integration step. See App. A.3 for function evaluation timings of each network.

In both panels of the figure, we see that the Euler solver has a notably poor efficiency in terms of function evaluations. This indicates that the velocity field learned by the shape network has non-trivial curvature. Considering the remaining solvers, the sample quality essentially saturates by $n_{\text{eval}} = 64$ and all non-Euler methods appear to have statistically-equal performance at this point. The bespoke samplers demonstrate the best retention in quality when looking toward smaller $n_{\text{eval}}$. In particular, the local BNS solver keeps an AUC below 0.6 for both classifiers even at 8 function evaluations. The global BNS solver achieves a large margin of improvement at $n_{\text{eval}} = 4$ for the high-level classifier. The local bespoke solver also shows an advantage in the high-quality regime. Specifically, its AUC is already saturated for both solvers at 32 function evaluations. As such, in a resource-limited scenario the efficiency gains offered by bespoke solvers can be translated into improved sample quality.

It is interesting to note that the performance of a given solver can be significantly different between high- and low-level classifiers. This is evident in the reversed rankings of, for example, the two bespoke solvers in each panel. The global BNS solver favors performance on the high-level classifier, while the local BNS solver is best on the low-level classifier. A similar exchange can be seen among the Midpoint and RK4 solvers, with the former being close to optimal at low level.

## 4.5 Performance

It is not trivial to test the overall performance of generative networks for calorimeter showers. In the previous sections we evaluated the networks using simple one-dimensional histograms, as in Figs. 5 and 7, or classifier AUC scores. A systematic approach to assess the quality of our generative networks, and a way to identify failure modes, is to examine the distribution of classifier predictions over the phase space or feature space $x$ [89]. A properly trained and calibrated classifier $C(x)$ learns the likelihood-ratio between the data and the generated distributions which, according to the Neyman-Pearson lemma, is the most powerful test statistic to discriminate between the two samples. This allows us to extract a correction weight over phase space

$$w(x) = \frac{C(x)}{1 - C(x)} \approx \frac{p_{\text{data}}}{p_{\text{model}}}(x) \,, \tag{21}$$

and to use the corresponding weight distributions as an evaluation metric. The weights have to be evaluated on the training data and on the generated data, because failure modes appear as tails in one of the two distributions [89]. To further analyze such failure modes, we can study showers with small or large weights as a function of phase space, using the interpretable nature of phase spaces in particle physics.

In Fig. 10 we show the classifier weights from the low-level classifier for DS2 and for DS3. We also include a table with the AUC scores of the high-level classifier trained on layer-wise features and the low-level classifier, where the ViT shows state-of-the-art results on DS2 and the high-level DS3. The peaks of the weight distributions are nicely centered around $w = 1$, symmetric towards small and large (logarithmic) classifiers, and show no significant difference between generated and training data. The weights for the networks encoding the full phase space and the latent diffusion are different, with a typical broadening of the distribution by a factor two around the peak and larger and less smooth tails. We still observe that the classifier misses the low-energetic noise affecting the sparsity and the voxel energy distributions. Despite the simple nature of the neural network, a sequence of fully connected layers, the

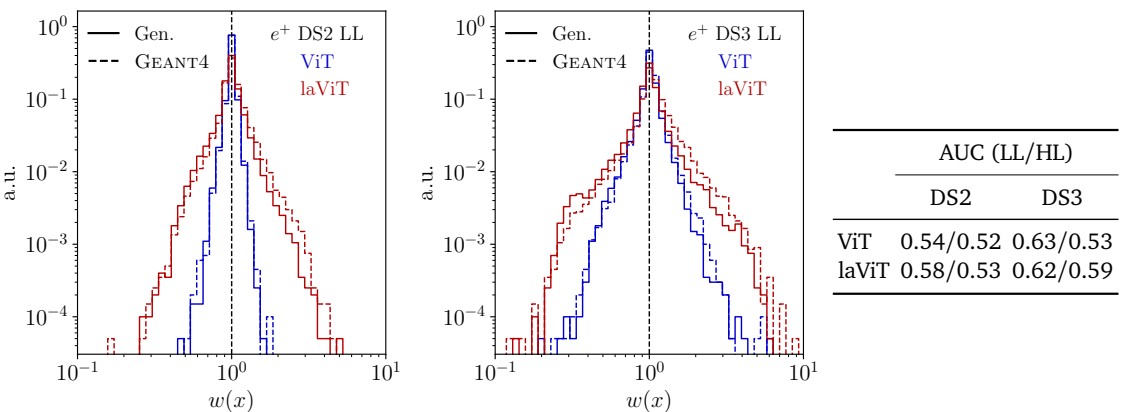

Figure 10: Learned low-level classifier weight distributions for DS2 (left) and DS3 (right). We compare the full-dimensional ViT and the latent laViT results and, for each of them, show weights for the generated sample and for a GEANT4 test sample. The table shows the AUC values for the trained classifier in each case.

main result from this performance test is that the classifier identifies additional failure modes related to the step from DS2 to DS3 and to the reduced latent space. We expect these failure modes correspond to cross-layer features, since we observe a correlation between the classifier weights and the shower depth introduced in Sec. 4.2, and the high-level AUC is similar across the two datasets. Details of the neural network classifier are listed in App. A.1.

## 5    Outlook

Calorimeter showers are one of the most exciting applications of modern generative networks in fundamental physics. Their specific challenge is the high dimensionality of the voxelized phase space, combined with extremely sparse data and an LHC-level precision requirement. In our case, the ML4Jets CaloChallenge datasets 2 and 3 include up to 40k dimensions for the target phase space.

In this situation, diffusion networks allow us to go a step beyond standard normalizing flows. Our CaloDREAM architecture first factorizes the generation of detector showers into an energy network and a shape network. Both networks are trained using Conditional Flow Matching. The former generates the layer energies using a transformer backbone with self-attention and cross-attention blocks. For the latter, we use a 3-dimensional vision transformer, operating on patches of the target phase space.

For DS2 this combination of networks is a safe architecture choice, in the sense that it can be trained without problems and reproduces all features, within layers and across layers, with high precision. We can use a VAE to reduce the dimensionality using latent diffusion. We find essentially no loss in performance, except for the reproduction of low-energy voxels and, with it, sparsity, which can be improved by introducing an MeV-level energy threshold. Because diffusion networks are slower than alternative generative networks, we use bespoke samplers to enhance their generation speed, at no cost of the precision and improving the fidelity in case of limited resources.

For DS3 the performance of the CaloDREAM generators remains qualitatively the same, but the shape network reaches the limit in terms of available computation time. This is a motivation to again employ latent diffusion. We find excellent performance of the latent diffusion

architecture; with the right choice of energy thresholds even the sparsity distribution is reproduced correctly. However, further studies are needed to understand the effects of mapping the distributions into real detectors with irregular geometries, more complex distributions from different incident particles, e.g. hadrons, and varying angle of impact.

Our study shows that modern generative networks can be used to describe calorimeter showers in highly granular calorimeters. When the number of phase space dimensions becomes very large and the data becomes sparse, a latent diffusion network combined with an (autoregressive) transformer and bespoke sampling provides excellent benchmarks in speed and in precision.

**Note added**   A potentially similar approach, CaloDiT, also using a diffusion transformer to tackle calorimeter showers has been shown at ACAT 2024.

# Acknowledgements

We would like to thank Theo Heimel, Lorenz Vogel, and Anja Butter for extremely helpful discussions. This research is supported through the KISS consortium (05D2022) funded by the German Federal Ministry of Education and Research BMBF in the ErUM-Data action plan, by the Deutsche Forschungsgemeinschaft (DFG, German Research Foundation) under grant 396021762 – TRR 257: *Particle Physics Phenomenology after the Higgs Discovery*, and through Germany's Excellence Strategy EXC 2181/1 – 390900948 (the *Heidelberg STRUCTURES Excellence Cluster*). SPS is supported by the BMBF Junior Group Generative Precision Networks for Particle Physics (DLR 01IS22079). The authors acknowledge support by the state of Baden-Württemberg through bwHPC and the German Research Foundation (DFG) through grant no INST 39/963-1 FUGG (bwForCluster NEMO).

# A Further details

## A.1 Hyperparameters

| Parameter | DS2 & DS3 |
|---|---|
| Epochs | 500 |
| LR sched. | cosine |
| Max LR | $10^{-3}$ |
| Batch size | 4096 |
| ODE solver | Runge-Kutta 4 (50 steps) |
| Network | transformer |
| Dim embedding | 64 |
| Intermediate dim | 1024 |
| Num heads | 4 |
| Num layers | 4 |
| Network | dense feed-forward |
| Intermediate dim | 256 |
| Num layers | 8 |
| Activation | SiLU |

Table 1: Parameters for the autoregressive energy network in Sec. 3.1.

| | ViT | | laViT | |
|---|---|---|---|---|
| Parameter | DS2 | DS3 | DS2 | DS3 |
| Patch size | (3, 16, 1) | (3, 5, 2) | (3, 1, 1) | (3, 2, 2) |
| Embedding dimension | 480 | 240 | 240 | 240 |
| Attention heads | 6 | 6 | 6 | 6 |
| MLP hidden dimension | 1920 | 720 | 960 | 960 |
| Blocks | 6 | 6 | 10 | 10 |
| epochs | 800 | 600 | 800 | 400 |
| batch size | 64 | 64 | 128 | 128 |
| LR sched. | | cosine | | |
| Max LR | | $10^{-3}$ | | |
| ODE solver | | Runge-Kutta 4 (20 steps) | | |

Table 2: Parameters for the shape networks in Sec. 3.2, for the full and the latent space.

| Parameter | Value |
|---|---|
| Optimizer | Adam |
| Learning rate | $2 \cdot 10^{-4}$ |
| LR schedule | reduce on plateau |
| Decay factor | 0.05 |
| Decay patience (epochs) | 20 |
| Batch size | 1000 |
| Epochs | 200 |
| Number of layers | 3 |
| Hidden nodes | 512 |
| Dropout | 20% |
| Activation function | leaky ReLU |
| Training samples | 70k |
| Validation samples | 10k |
| Testing samples | 20k |

Table 3: Parameters for the classifier network used to calculate the weights of Fig. 10.

| Parameter | Value | |
|---|---|---|
| | DS2 | DS3 |
| Loss | BCE + $\beta$KL | |
| $\beta$ | $10^{-6}$ | |
| Epochs | 200 | |
| Out activation | sigmoid | |
| Lr sched. | OneCycle | |
| Max lr | $10^{-3}$ | |
| # of blocks | 2 (+ bottleneck) | |
| Channels | (64, 64, 2) | |
| Dim. bottleneck | (2, 15, 9, 9) | (2, 9, 26, 16) |
| Kernels | [(3,2,1), (1,1,1)] | [(5,2,3), (1,1,1)] |
| Strides | [(3,2,1), (1,1,1)] | [(2,2,1), (1,1,1)] |
| Paddings | [(0,1,0), (0,0,0)] | [(0,1,0), (0,0,0)] |
| Normalized cut | $1 \cdot 10^{-6}$ | |

Table 4: Parameters of the autoencoder for DS2 and DS3 used for the laViT network in Sec. 3.3.

| Parameter | Value |
|---|---|
| Reference solver | midpoint (100 steps) |
| Initialization | Euler |
| Optimizer | Adam |
| Learning rate | $1 \cdot 10^{-3}$ |
| Batch size | 100 |
| Max iterations | 5000 |
| Stopping patience (iterations) | 200 |

Table 5: Parameters used to train BNS solvers, described in Sec. 3.4.

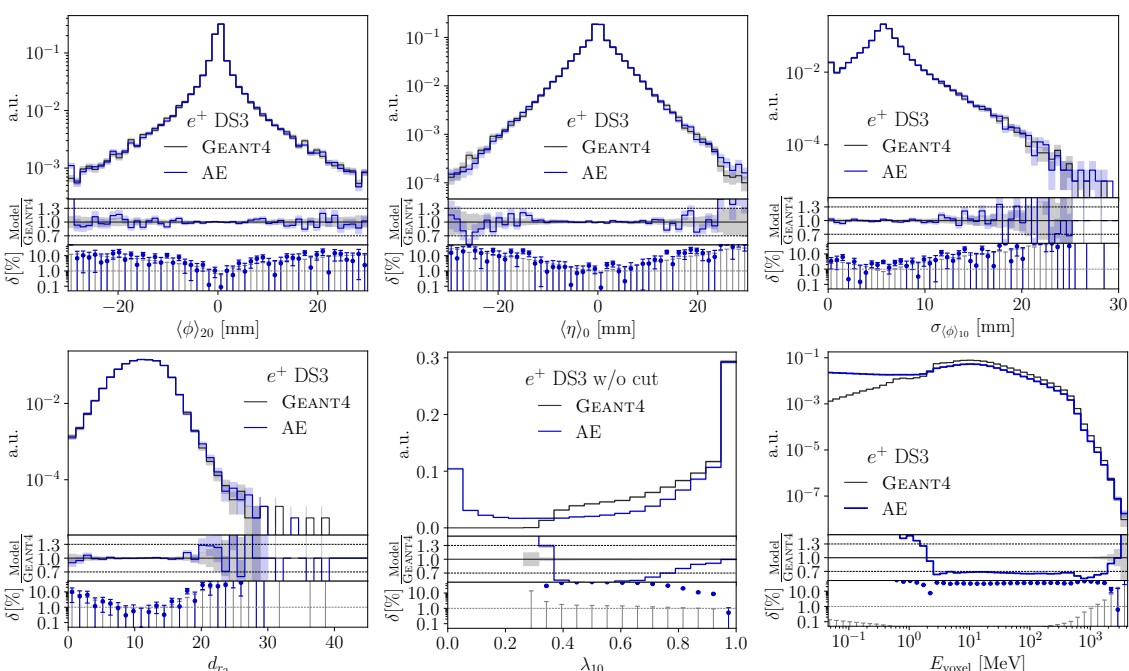

Figure 11: Selection of high-level features sensitive to the reconstruction of the autoencoder for DS3.

## A.2 Autoencoder

The VAE introduced in Sec. 3.3 is trained separately, using the BCE reconstruction loss

$$\mathcal{L}_{\text{VAE}} = -\left\langle x \log(x_\psi) + (1-x) \log(1-x_\psi) \right\rangle_{p_\psi(r|x)} + \beta D_{\text{KL}}[p_\psi(r|x), \mathcal{N}(0,1)] . \tag{22}$$

This loss provides notably better reconstruction quality than the standard MSE loss, both in terms of high-level features and a neural network classifier trained to distinguish reconstructed showers from an independent test set. A detailed description of the network architecture is provided in Tab. 4. Each block consists of three Conv2d operations that preserve the number of channels of which the final one downsamples according to the stride and padding parameters. In addition, we break the translation equivariance by adding the coordinates of each input to the activation map as new channels [56].

In Fig. 11 we provide a set of kinematic distributions similar to Fig. 7 for DS3, to illustrate the VAE reconstruction. We find that the only missing feature in the learned manifold is the distribution of the low-energetic voxels, also reflected in the sparsity. We also train a classifier using the hyperparmeters of Tab. 3 on the low-level features which gives an AUC score of 0.512(5) consistently for both, DS2 and DS3.

## A.3 Timing

In Sec. 4.4, we study the sampling cost of networks in terms of the number of function evaluations $n_{\text{eval}}$. Here we provide timing measurements for a single forward pass of each of our CFM networks, using a batch size 100. We ran tests using a single NVIDIA H100 GPU and summarize the results in Tab. 6. The times for the energy network are identical across the two datasets since there is no change in the network architecture. Also note that since the energy model is autoregressive, sampling with an $N$-step solver uses $N \times L$ function evaluations, where $L$ is the number of calorimeter layers.

| Network | Time (ms) | |
|---|---|---|
| | DS2 | DS3 |
| Energy | 0.37±0.01 | 0.37±0.01 |
| Shape (ViT) | 17±1 | 84±8 |
| Shape (LaViT) | 31±1 | 63±6 |

Table 6: Timings for one network forward pass using batch size 100 on an NVIDIA H100.

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
