# Peer review of "CaloDREAM -- Detector Response Emulation via Attentive flow Matching"

_SciPost Physics_

## Round 2 · Referee Report · Anonymous (Referee 1) · 2024-7-5

Strengths

See report below.

Weaknesses

See report below.

Report

Simulating particle material interaction in high energy physics detector is a time/CPU consuming task, especially the interaction in the calorimeters. Generative machine learning techniques provide a new approach for this task, with the current challenge of reaching high dimensionality and high fidelity. The community released common datasets via the CaloChallenge. This work reported the "CaloDREAM" method and tested it on the dataset 2 and 3 provided by the CaloChallenge.

The method is a novel method using various techniques. It decouples the deposited energy per layer from the shower shape, and use independent energy network and shape network to generate both separately. The energy network uses a transfusion technique and the shape network uses a vision transformer.

Regarding the results, the networks are still struggling to learn some details of the calorimeter showers, as can been seen from Fig 5-7, 11. Further refinement of the network and/or better training strategy would be needed.

The manuscript is well written and complete: it motivates the problem to solve in the field, and explain how these networks are constructed, followed by the results using the dataset 2 and 3 of the CaloChallenge, and finally closed with an discussion of the future research directions. I suggest to accept the paper with minor edition changes to address my comments/suggestions detailed below.

Requested changes

  1. Section 4.1 "obtaining AUC scores around 0.51": The AUC value is impressive, however, from Fig 4, especially u_44 distribution, the difference between G4 and CaloDREAM is quite high. Is it because the classifier is mainly affected by the peak of the distribution where the agreement looks better? Could you clarify in the text this AUC is training or test AUC?

  2. Section 4.3 "networks developed for lower-dimensional phase spaces also give the necessary precision for high-dimensional phase spaces": From the context it is not clear the motivation to maintain this assumption. Do you imply some degree of transfer learning? If so, it would be nice to detail the motivation.

  3. Section 4.4 "It is interesting to note that the performance of a given solver can be significantly different between high- and low-level classifiers." Not sure how the conclusion "significant difference" is drawn. From Fig 9, they are close to each other except for Euler. I suggest to rephrase this sentence to remove "significantly".

  4. Fig 10: Each plot show two sets of results - Geant4 and Gen. I don't get how Geant4 are obtained. I thought the plot shows the "weight" of each event that maps from Gen to Geant4, namely, in Eq 21, p_data is Geant4 distribution, p_model is Gen distribution. Am I missing something?

  5. Fig 10 table on the right: I don't find definitions for LL/HL.

  6. Fig 10 table on the right: These AUCs are all nice but DS3 HL are quite different in ViT and davit. Is it subject to large uncertainty? Did you get stable results by training the classifier a second time?

  7. Sect 5 "However, further studies are needed to understand the effects of mapping the distributions into real detectors with irregular geometries": the rest part in this paragraph is true for both datasets. Suggest to make a new paragraph.

  8. Appendix A.2 "which gives an AUC score of 0.512(5)": Again from Fig 11, the separation is quite large. I am not sure why the classifier gives such good AUC scores. Is this training or test AUC? Did you check the classifier score distribution?

  9. Table 6 worth commenting why LaTiV is slower on DS2 (assuming it should be faster?)

  10. Section 3.3 "Finally, the phase space configurations are provided by the the ...": duplicated "the"

  11. Section 3.4 "... with coupling layers [90] which stems from ...": reference of "which" is not clear.

Recommendation

Publish (easily meets expectations and criteria for this Journal; among top 50%)

  • validity: high
  • significance: good
  • originality: high
  • clarity: high
  • formatting: good
  • grammar: good

Author:  Ayodele Ore  on 2025-01-01  [id 5076]

(in reply to Report 1 on 2024-07-05)

We thank the referee for their comments, which we address in turn below.

  1. It is true that in Fig. 4, the precision is worst for the u_44 distribution. Since this disagreement lies in a low density region (note the logarithmic scale), its contribution to the AUC is indeed small. All AUC scores reported in the paper are on the testing set. We have clarified this in the text.

  2. We see how this sentence could be misleading. We meant to convey that the machine learning architectures used in the HEP ML community are primarily applied to lower dimensional problems compared to the DS3 dimensionality. We therefore want to test whether those architectures are applicable to high-dimensional problems as well. There is no degree of transfer learning implied. We adapted the statement by changing “networks developed for” to “architectures that have been successful on“.

  3. Here we are comparing the behavior of a given solver across the two panels. For example, at n_eval=8 the global bespoke solver has a high-level AUC of ~0.55 but a low-level AUC of ~0.7, with uncertainties less than 0.01. Similarly, the midpoint solver has high- and low-level AUCs of 0.6 and 0.65 respectively.

  4. The classifier is trained on Geant4 vs Gen. Once trained, we evaluate it on Gen as well as on Geant4, leading to two sets of weights. In theory the reciprocal of the “Geant4” weights shown in the plot should map from Geant4 to Gen, which is of course of no interest. However, by looking at the Geant4 weights in the plots, we can ensure that the classifier learned the likelihood ratio correctly. If we only look at the Gen weights, we may not identify cases where the generator suffers from mode collapse (i.e. if the Gen and Geant4 distributions have different support).

  5. We have added definitions for LL/HL to the text.

  6. Indeed, all AUC scores are stable up to the decimal places shown. We train 10 classifiers with different initializations and report the mean AUC from these runs. We have also added the error.

  7. We reworded this statement to clarify that it applies to both datasets.

  8. In Fig 11, the generator is mainly discrepant from Geant4 in sparsity and voxel energy. These are related failure modes; the autoencoder tends to fill voxels with extremely low-energy values rather than producing exact zeros. However, the classifier for which we report the AUC is a low-level classifier, trained on the voxels themselves. Therefore it does not directly ‘see’ the sparsity feature shown in Fig 11. In practice, it is difficult for the classifier to distinguish exact zeros from very small values. The low AUC is in general agreement with the perfect reconstruction of the physical features (shower centrality and width) by the autoencoder. Again, the AUC is always calculated on the test dataset.

  9. Our main motivation to use a latent generative model is to tackle the high dimensionality of DS3. There, we find that an autoencoder with latent space compression factor of 2.5 is a suitable choice. For consistency, we also use the same reduction factor for DS2. However, the dimension of DS2 is of course closer to the intrinsic dimensionality of the showers. As such, one can expect that the DS2 autoencoder latent space has a complicated structure. In practice, this means we need to run a more precise (and therefore slower) diffusion model in order to generate high quality showers. We added a sentence in section 3.3, when introducing the reduction factor F, and clarified in the timing related appendix that no extensive hyperparameter scan has been performed.

  10. Typo fixed.

  11. Fixed ambiguity by splitting the sentence in two.

---

## Round 2 · Referee Report · Anonymous (Referee 2) · 2024-7-16

Report

The accurate an efficient simulation of calorimeter showers is a topic of high interest in collider physics. Currently, traditional calorimeter simulation are the major bottleneck of the Large Hadron Collider (LHC) simulation pipeline. The topic covered in the manuscript is thus of major significance.

The authors propose a novel and highly promising machine learning based method, dubbed CaloDREAM, capable of generating high-fidelity showers. Their strategy involves several state-of-the art machine learning techniques: Conditional Flow Matching (CFM) autoregressive Vision Transformers (ViT) and Latent diffusion with Variational Auto-encoders (VAE). The time consuming step coming from the CFM implementation is alleviated using bespoke samplers for which they compare several methods. Their implementation is trained and tested with datasets 2 and 3 from the CaloChallenge and the accuracy is assessed via classifier-based methods.
The manuscript is well written. It provides a, generally, well explained description of their methods, and potential ways it could be further improved.

Requested changes

-If I understand correctly, you used the same classifier architecture for all accuracy tests, not only for the ones on Fig. 10. This should be clarified in the manuscript. Also, how were these parameters chosen? Is it clear that they are optimal for both the high level and low level tests?

  • You feed the vision transformer patches made of n voxels. After the ViT generation how are the patches transformed back into voxels, is the energy per patch uniformly distributed over the n voxels? Have you checked if there is a noticeable error/loss of information just from mapping voxels to batches and back to voxels?

-According to table 6 shape sampling DS2 is faster with ViT than with LaViT, why is this the case?

-Typo at ‘can be be understood ’ in the introduction.

Recommendation

Ask for minor revision

  • validity: high
  • significance: high
  • originality: high
  • clarity: high
  • formatting: excellent
  • grammar: excellent

Author:  Ayodele Ore  on 2025-01-02  [id 5078]

(in reply to Report 4 on 2024-07-16)

We thank the referee for their comments, which we address in turn below.

-Regarding classifier architectures: All classifiers use the same architecture as prescribed by the Calo Challenge. For the weight distributions in Fig. 10, we improve the calibration by introducing weight decay, dropout, and early stopping. This reduces the overfitting observed while running the standard CaloChallenge pipeline for both DS2 and DS3. We updated the caption of Table 3 to clarify this.

-Regarding transformer patching: After segmenting the calorimeter, each patch is embedded into a vector with more dimensions than the number of pixels in the patch. For example, the largest patch size we use is (3, 16, 1) ~ 48 pixels and we use a 480-dim embedding space such that there is no information bottleneck. The transformer output includes a projection from the embedding size to a vector with as many dimensions as there are pixels in a patch (e.g. 480 -> 48 in the example above). At this point, the values are interpreted as (normalized) energy values for each pixel.

-Regarding sampling time: [Response copied from point 9 in Report #1] Our main motivation to use a latent generative model is to tackle the high dimensionality of DS3. There, we find that an autoencoder with latent space compression factor of 2.5 is a suitable choice. For consistency, we also use the same reduction factor for DS2. However, the dimension of DS2 is of course closer to the intrinsic dimensionality of the showers. As such, one can expect that the DS2 autoencoder latent space has a complicated structure. In practice, this means we need to run a more precise (and therefore slower) diffusion model in order to generate high quality showers. We added a sentence in section 3.3. when introducing the reduction factor F and clarified also in the timing related appendix that no extensive hyperparameter scan has been performed.

-Typo fixed.

---

## Round 2 · Referee Report · Anonymous (Referee 3) · 2024-7-16

Strengths

  1. High quality output
  2. Adoption of new, cutting-edge techniques
  3. Focus on reducing inference time

Weaknesses

  1. Tersely written, especially when describing techniques, requiring very careful reading to understand the exact procedures
  2. Some aspects of the results are not fully quantified (missing are e.g. measurements of inference time, more thorough characterization of physics performance)

Report

This paper presents a model competitive with other leading entries in the community challenge. It is based on new techniques, some of which may be adopted more broadly now that they have been demonstrated to work. I recommend accepting the paper after some minor improvements to clarify details in the text, as suggested below.

Requested changes

  1. page 1, par 1: The claim that simulating detector interaction is a bottleneck in speed and precision would be more convincing if it were supported by some citations. During Run 2, the HEP Software Foundation led an effort to quantify this, which can be found in https://arxiv.org/abs/1803.04165.
  2. page 1, par 2: The assertion "we have to ensure that LHC simulations remain first-principle and are not replaced by data-driven modelling" should be supported by further discussion. One could argue that, depending on the hypothesis being tested, only event generation necessarily needs to occur from first principles.
  3. page 1, par 4: Why is CaloChallenge dataset 1 omitted? It includes some interesting distinctions compared to the other datasets: a different geometry and different particle types. This paper would certainly be improved by including it, but if that is not feasible, at least some discussion should be added to say why it is not there.
  4. page 2, footnote: typo avaiable -> available
  5. page 5, par 1 / fig 1: If the Transfusion encoder/decoder is trained along with the CFM generator (rather than using a frozen, pretrained encoder/decoder), does that imply that the outputs c_i implicitly depend on the diffusion step time t, and if so, would it be better to condition these outputs on t explicitly?
  6. page 6, par 3: This discussion of patching does not specify how the actual patch sizes used in Table 2 were chosen. Was a hyperparameter scan performed?
  7. page 9, par 2: Is it just the a_i and b_i parameters that are learnable when training the bespoke solver? This is not quite clear as currently written.
  8. fig 6: Though the 1 MeV cut decreases the proportion of events populating the first sparsity bin, the overall agreement in the sparsity distribution actually worsens. Is this understood?
  9. page 13, par 2: This paragraph may be intended to explain the question posed above. Did the authors mean to say that it fixes the sparsity down to λ_10 <~ 0.7, rather than up to? Otherwise it does not make sense.
  10. section 4.5: While the classifier AUC and other related quantities are powerful, they also suffer from some limitations and caveats (assumption of optimality in training, dependence on amount of training data, etc.). The paper would benefit from an additional non-parametric evaluation using e.g. the Frechet Particle Distance.
  11. page 19, par 2: typo hyperparmeters -> hyperparameters

Recommendation

Ask for minor revision

  • validity: high
  • significance: high
  • originality: high
  • clarity: good
  • formatting: good
  • grammar: good

Author:  Ayodele Ore  on 2025-01-02  [id 5077]

(in reply to Report 3 on 2024-07-16)

We thank the referee for their comments, which we address in turn below.

  1. We agree and have added relevant citations.

  2. We agree that this point potentially warrants its own discussion, though we do not wish to detract from the main ideas of the paper. We reworded this statement as “we must avoid the case where effects of interest are absorbed into LHC simulations as a result of data-driven modelling.”, which we hope accommodates the referee’s point.

  3. It is not due to any technical limitations that we omitted dataset 1. Rather, it has previously been shown that normalizing flows can give precise generation in dataset 1 (Ernst et al. 2312.09290), while being much faster than diffusion approaches. In this paper we are focused in particular on the challenge of large-dimensional datasets, where normalizing flows struggle in terms of trading off quality and computational cost. We added a sentence to clarify this.

  4. Typo fixed.

  5. The transfusion encoder/decoder is indeed trained along with the CFM generator. However, the outputs c_i do not depend on the diffusion step time t. Here, the transformer functions as a preprocessing step: it embeds the necessary information of the overall condition (E_inc) and all previously generated u’s, which is intrinsically time-independent. Inputting the time t to the transformer would break this time-independence. This would imply that the network has to be autoregressive not only at time t=0 but at any time t, which is a much harder training task. Importantly, the transformer would also have to be evaluated at every time t, which would increase the computational cost of sampling immensely.

  6. We did not systematically scan the patch size. The final values are the result of some manual tweaking and could likely be improved. We added a comment stating this explicitly.

  7. In addition to the a_i and b_i, the time discretization is also learned (the t_i with 0 < i < N). We have clarified in the text that these are the only trained parameters.

  8. This is a combined effect of the thresholding and the voxel normalization. The level at which the autoencoder fails is fixed (e.g. 1.e-4) in the normalized space. However, after undoing the normalization, each layer in each shower has a different corresponding energy threshold. The cut we apply will first adjust layers with little energy deposition. For larger total energy deposits, the energy threshold is not large enough to match the autoencoder and it will not largely affect the first bin in the sparsity. In agreement with this, we observe that the sparsity in the first layer improves (in terms of chi squared) while it degrades for layers with large typical energy deposition (e.g. layer 20). It is not obvious how to distribute the energy from a 6000d space such that the contribution to the other bins is uniform. As a comment: This is something we already touch on in the text under the “DS3 showers” section.

  9. Your understanding is correct, and we have fixed this.

  10. A detailed comparison of generative models for calorimeter showers (including CaloDREAM) has been conducted in a community paper (Krause et al. 2410.21611). There, the FPD and a number of other metrics are evaluated on CaloDREAM samples. We have added a citation to this paper.

  11. Typo fixed.

---

## Round 3 · Referee Report · Anonymous (Referee 1) · 2025-1-10

Strengths
Weaknesses
Report
Requested changes
-
reply to " Here we are comparing the behavior of a given solver across the two panels. For example, at n_eval=8 the global bespoke solver has a high-level AUC of ~0.55 but a low-level AUC of ~0.7, with uncertainties less than 0.01. Similarly, the midpoint solver has high- and low-level AUCs of 0.6 and 0.65 respectively." Thanks for the explanation. It is clear to me now. I would suggest to incorporate a brief explanation around that line to guide readers.
-
reply to "The classifier is trained on Geant4 vs Gen. Once trained, we evaluate it on Gen as well as on Geant4, leading to two sets of weights. In theory the reciprocal of the “Geant4” weights shown in the plot should map from Geant4 to Gen, which is of course of no interest. However, by looking at the Geant4 weights in the plots, we can ensure that the classifier learned the likelihood ratio correctly. If we only look at the Gen weights, we may not identify cases where the generator suffers from mode collapse (i.e. if the Gen and Geant4 distributions have different support)." Same for this one to avoid potential confusion as I had.
Recommendation
Ask for minor revision

---

## Editorial Decision

unknown